

# Dinocyst assemblage constraints on oceanographic and atmospheric processes in the Eastern Equatorial Atlantic over the last 44 ka

**Hardy William** [a] *, **Penaud Aurélie** [a] *, **Marret Fabienne** [b], **Bayon Germain** [c], **Marsset Tania** [c], **Droz Laurence** [a]

*(a) UMR 6538 Domaines Océaniques, IUEM-UBO, F-29280 Plouzané, France.*

*(b) School of Environmental Sciences, University of Liverpool, Liverpool, L69 7ZT, UK*

*(c) IFREMER, UR Géosciences Marines, BP 70-29280 Plouzané, France*

*Corresponding author. Tel.: +33-298-498-741; fax: +33-298-498-760

*E-mail address:* william.hardy@univ-brest.fr



**ABSTRACT**
A new 44 ky-long record of dinoflagellate (phytoplanktonic organisms) cysts (dinocysts) is
presented from a marine sediment core collected on the Congolese margin with the aim to
reconstruct past hydrological changes of the Equatorial Eastern Atlantic Ocean since Marine
Isotopic Stage 3. Our high-resolution dinocyst record indicates that significant temperature
and moisture variations occurred across the glacial period, the last deglaciation and the
Holocene. The use of specific dinocyst taxa, indicative of fluvial, upwelling and Benguela
Current past environments for instance, provides insights into the main forcing mechanisms
controlling paleohydrological changes at orbital timescales. In particular, we are able, for the
last 44 ky to correlate fluvial-sensitive taxa to monsoonal mechanisms related to precession
minima/obliquity maxima combinations. While upwelling mechanisms appear as the main
driver for dinoflagellate productivity during MIS 2, dissolved nutrient-enriched Congo River
inputs to the ocean also played a significant role in promoting dinoflagellate productivity
between approximately 15.5 and 5 ka BP. Finally, this high resolution dinocyst study permits
to precisely investigate the sub-orbital timing of the last glacial-interglacial termination
including an atypical warm and wet oceanic LGM signature, northern high latitude abrupt
climate change impacts in the Equatorial Eastern Atlantic, as well as a two-steps mitigation of
moisture conditions during the Holocene at around 7-6 and 4-3.5 ka BP.

35 .

*KEYWORDS: Dinoflagellate cysts; Congolese margin; Deglaciation; Holocene;*
*Paleoproductivity; Monsoon dynamic*



**1. INTRODUCTION**
Reconstructions of Late Quaternary and Holocene paleoceanographic changes at the Western
African margin and associated Benguela upwelling system have identified orbital and sub-
orbital controls on sea-surface and continental environmental conditions (Holzwarth et al.,
2007). More specifically, several palynological studies carried out in the Equatorial Eastern
Atlantic Ocean, combining analysis of pollen grains and cysts of dinoflagellates (dinocysts),
have provided a wealth of information on land-sea interactions in the intertropical region,
through investigation of sea-surface and terrestrial vegetation changes over the last climatic
cycles (Shi et al., 1998; Marret and Zonneveld, 2003; Dupont and Behling, 2006; Marret et
al., 2008; Kim et al., 2010; Bouimetarhan et al., 2012; Marret et al., 2013). However, these
above-mentioned studies mainly focused on the comparison between periods of extreme
climatic conditions, such as the Last Glacial Maximum (Mix et al., 2001) and the Holocene
Climatic Optimum, showing that higher primary productivity conditions occurred during
glacial periods in response to an increase of upwelling activity while enhanced freshwater
discharges from the continent occurred during interglacials. In comparison, the Last
Deglaciation period, which consisted in a shift from upwelled cold waters (associated with
dry conditions on land - glacial) to monsoonal regimes (associated with warm waters offshore
- interglacial), has been less studied and its timing in this area remains poorly defined, mainly
due to a lack of high-resolution investigations.
In this study, we have investigated a marine sediment core (KZAI-01) recovered during the
ZaiAngo I cruise (Savoye et al., 2000) at the West-African continental slope, upstream the
Congo deep-sea fan. This core is characterized by a high average sedimentation rate (about 25
cm/ka, maximum of 50 cm/ka) that enables to provide high-resolution paleoenvironmental
records for the last 44 ka (Bayon et al., 2012). We have combined new dinocyst analysis with
a set of already published geochemical data for sediment provenance and weathering proxies





(Bayon et al., 2012). The comparison between terrestrial and marine proxy data can then be
used to discuss about the links between environmental changes that have occurred in the
Congo catchment area and past sea-surface oceanographic through dinoflagellate productivity
variations.
Several objectives have motivated this study:
1) To document the potential of dinocysts for reconstructing sea-surface environments in

71       the Eastern Atlantic Ocean and discussing about the links between continental and

72       hydrological changes over the last 44 ka,

2) To discuss orbital forcing impacts in our recorded dinocyst observations and the

74       potential influence of the monsoonal activity on sea-surface past conditions,

3) To precisely characterize, in the Equatorial Atlantic Ocean, the timing of the Last

76       Deglaciation at a millennial time-scale resolution.


**2. ENVIRONMENTAL CONTEXT ON THE CONGOLESE MARGIN**
The Congo River drains the second catchment area of the world with a total surface of
3,600,000 km² and a mean flow of 41,000 m³/s. This river feeds the Congo deep-sea fan
(Babonneau et al., 2002; Droz et al., 2003; Savoye et al., 2009; Picot et al., 2016), one of the
largest deep-sea fans in the world, *via* a submarine Pliocene canyon (Anka et al., 2009) still
active at present (Heezen and Hollister, 1964; Khripounoff et al., 2003)
*2.1. Present-day atmospheric context*
Climatic patterns in the Congo Basin are controlled by the seasonal latitudinal migration of
the Tropical Rainbelt (TR, Fig. 1), which is associated offshore with high sea-surface
temperatures (SST) and low salinities (Zarriess and Mackensen, 2010; Arbuszewski et al.,
2013). This low pressure belt is characterized by moist air ascension and large tropical
rainstorms, generated by the association of the Tropical and African Easterly Jets in the



Northern Hemisphere (Nicholson, 2009). The TR and the Inter Tropical Convergence Zone
(ITCZ) constitute the complex convective system of African monsoon. which shift seasonally
from a northward position during boreal summer to a southward position during boreal winter
(Hsu and Wallace, 1976). While the central part of the Congo Basin is characterized by an
equatorial regime, its northern and southern parts alternate between wet and dry seasons
(Prance, 1984; Leroux, 2001). This results in a latitudinal distribution of the vegetation from
rainforests to savannahs across the whole Basin (Prance, 1984). Easterly winds from the
Indian Ocean also brings moisture to the Congo Basin, in particular during the austral
summer, due to the presence of the Congo Air Boundary convergence zone (CAB, Tierney et
al., 2011), also evidencing the influence of the eastern African monsoon system in central
Africa.

***2.2. Present-day oceanographic context***
Surface water masses from the Congolese margin are largely influenced by the Angola
Current (AC; Figure 1), a clockwise subequatorial gyre located above the north-eastern part of
the Subtropical Gyre (Lass and Mohrholz, 2008). The warm waters of the AC meet the cool
waters of the couple Benguela Current and Coastal Benguela Current (BC and cBC; Figure 1)
at around 16°S at the Angola-Benguela Front (Lass and Mohrholz, 2008). This cool surface
current causes weak evaporation and aridity conditions on the adjacent continent (Gordon et
al., 1995), as well as water mass stratification on the continental shelf, itself depleted in
oxygen (Gordon et al., 1995).
The South Atlantic Anticyclone, driving the Subtropical Gyre, generates SE trade winds on
the SW African margin, and consequently upwelling cells throughout the BC (Gordon et al.,
1995; Lass and Mohrholz, 2008). These upwelled waters bring deep nutrient-rich waters that
promote high primary productivity in surface waters. The Benguela upwelling system is



limited northward around the ABF location (Jansen et al., 1996; Lass and Mohrholz, 2008).
Congo River freshwater discharges also exert an influence on the regional oceanographic
setting, in particular because of the relative weakness of the Coriolis force near the Equator
that allows river plumes to extend far from the coast (da Cunha and Buitenhuis, 2013). This
mechanism also contributes to promote fluvial upwelling and thus to additional nutrients
exported to surface waters. Today, rainforests prevent soils from active erosion and therefore
prevent the delivery of substantial fluvial nutrient supplies to the Gulf of Guinea.

**3. MATERIAL AND METHODS**
*3.1. Stratigraphy of core KZAI-01*
Core KZAI-01 (5°42.19'S; 11°14.01'E; 816 m water depth; 10.05 m length; Figure 1) was
recovered during the 1998 ZaiAngo 1 cruise aboard the *Atalante* (Savoye, 1998).
First published age model of core KZAI-01 (Bayon et al., 2012) was derived from (Table 1)
seven AMS [14]C dates on carbonates (bulk planktonic foraminifera or mixed marine
carbonates), and two age constraints obtained by tuning core KZAI-01 to core GeoB6518-1
(well-dated sedimentary record from the nearby area; Figure 1) (Bayon et al., 2012).
In this study, three new AMS [14]C dates on carbonates have been added between 370 and 620
cm so as to obtain a more robust stratigraphy for the Last Glacial period (Table 1; Figure 2).
We have also added three new age constraints obtained by tuning core KZAI-01 to well dated
nearby core GeoB6518-1 (AMS [14]C dates on monospecific foraminifera; Schefuss et al.,
2005; Figure 2). This enables us to strengthen the chrono-stratigraphy of the study core for the
Early Holocene as well as for the base of KZAI-01 core, not constrained by AMS [14]C dates
below 851 cm.
All radiocarbon dates were calibrated to calendar ages with the 7.0 Calib program associated
with a 400 years correction for the marine age reservoir (Minze Stuiver, 1992; Reimer, 2013),



and the final age model was built through linear regression between all stratigraphic pointers
(cf. Table 1; Figure 2). Mean calculated sedimentation rates are around 25 cm/ky.

***3.2. Palynological analysis***
*3.2.1. Laboratory procedure for dinocyst extraction*
In this study, 203 samples were analysed for the period covering the last 44 ka with a 5 cm
sampling interval (mean resolution analysis of about 200 years throughout the core, ranging
between 20 and 800 years, according to the established age model). The preparation technique
for palynological analysis followed the procedure described in Marret et al. (2008). Calibrated
tablets of known concentrations of *Lycopodium* spores were added in each sample before
chemical treatments in order to estimate palynomorph concentrations (number of
dinocysts/cm$^3$ of dry sediments), and chemical and physical treatments included cold HCl
(10%), cold HF (40%), and sieving through a single use 10 µm nylon mesh screen. The final
residue was mounted between slide and coverslip with glycerine jelly coloured with fuschin.
When the recommended number of 300 dinocysts could not be reached, a minimum of 100
specimens was counted on each sample (F Fatela, 2002), using a Leica DM 2500 microscope
at ×630 magnification. Fifteen samples, containing less than 100 specimens, were excluded
from the dinocyst results. Dinocyst concentrations were based on the marker grain method (de
Vernal et al., 1999) and dinocyst assemblages were described by the percentages of each
species calculated on the basis of the total dinocyst sum including unidentified taxa and
excluding pre-Quaternary specimens. In addition to dinocyst counts, freshwater microalgae
*Pediastrum* and *Concentricystes* were also identified and counted so as to discuss river
discharge intensifications in parallel with our dinocyst data.




*3.2.2. Dinocysts as potential witnesses for past primary productivity changes*
Paleoproductivity regimes in the Equatorial Ocean can be inferred from our fossil
assemblages thanks to the transfer function based on the Modern Analogue Technique (MAT;
Guiot and de Vernal, 2007) developed for the Tropical Atlantic Ocean (n=208 modern
analogues; Marret et al., 2008). Mean annual Primary Productivity (PROD_Modis, Radi et al.,
2008 and PROA, Antoine et al., 1996) can then be quantified with a prediction error of 65,07
g/m². However, results issued from these quantifications include a number of limitations and
criticisms that will be discussed later in a paper devoted to primary productivity regimes in
the study area with an extended space vision and a data-model inter-comparison approach. In
this paper, we only focus on the dinoflagellate phytoplanktonic compartment through past
dinocyst specific observations. Indeed, among dinocyst assemblages, it is possible
distinguishing between cysts formed by dinoflagellates with a strict nutritional strategy based
on heterotrophy that we will refer as "heterotrophic cysts", and other cysts formed by
dinoflagellates for which the nutritional strategy can be complex involving either autotrophy,
heterotrophy or mixotrophy and that we will refer as "non heterotrophic cysts". It is well
known that relative abundances of total heterotrophic cysts can be used as a signal for
dinoflagellate primary productivity, and indirectly for marine productivity, considering that
heterotrophic dinoflagellates mainly feed on marine micro-organisms including other
dinoflagellates (whatever their nutritional strategies), diatoms and other micro-algae (e.g
Zonneveld et al., 2013).

**4. DINOCYST RESULTS ON CORE KZAI-01**
*4.1 Dinocyst concentrations*
A total of 53 different dinocyst taxa (Annexe 1) have been identified in the studied samples,
with an average of 15 different taxa for each sample (Figure 3). Total dinocyst absolute



concentrations in sediments are generally very low, from 100 cysts/cm$^3$ to 12,000 cysts/cm$^3$
(Figure 3). These low total concentrations in the study area are mainly attributed here to a
strong dilution of the organic matter by terrigenous inputs (cf. Figure 3 with the obvious
negative correlation between maximal values of terrestrial inputs (Ti/Ca and minimal values
of dinocyst concentrations), but also to a probable competition with diatom productivity
(Marret et al., 2008).
Higher total cyst concentrations are recorded between 850 and 450 cm (37.5 - 15.5 ka BP;
mean value of 3,000 cysts/cm$^3$), as well as between 90 and 30 cm (4 - 2.4 ka BP; mean value
of 6,000 cysts/cm$^3$), for which two maxima are observed with 10,900 and 11,200 cysts/cm$^3$,
respectively (Figure 3). Increases in total dinocyst concentrations can be mainly attributed to
increasing occurrences of *Operculodinium centrocarpum* or *Lingulodinium machaerophorum*
(Figure 3). Heterotrophic cyst concentrations (mainly led by *Brigantedinium* spp. and
*Echinidinium* species; Figure 3) as well as other cyst concentrations reach their maximal
values during the same main interval, i.e between 850 and 450 cm, but are three times lower
for heterotrophics (Figure 3). Also, higher total heterotrophic relative abundances, mainly
driven by *Brigantedinium* spp. percentages all along the record (Figure 3), as well as those of
*Echinidinium* spp. between 450 and 90 cm (15.5 - 4 ka BP; Figures 3 and 5), are strongly
correlated with lower total dinocyst concentrations, especially between 15 and 4 ka BP
(Figures 3 and 4). This could be consistent with the fact that diatoms, but also dinoflagellates,
consist in the main food for strict heterotrophic dinoflagellates (Marret and Zonneveld, 2003),
therefore echoing the previous idea of a competition between dinoflagellate and diatom
phytoplanktonic productivity in the study area (Marret and Zonneveld, 2003).
Even if heterotrophic dinocyst concentrations can firstly be attributed to
dilution/concentration processes in sediments, the transition between generally higher cyst
concentrations and lower ones observed at 450 cm (15.5 ka cal. BP) is synchronous with a



marked shift in biogenic silica (BiSiO$_2$) and total organic carbon (TOC) observed in a nearby
core (Schneider et al., 1997). This could lead us to suggest different marine productivity
patterns before and after 15.5 ka BP. Based at least on the fact that these data indicate
generally similar trends, an atypical pattern is however obvious at 90 cm. While heterotrophic
concentrations remain low, and despite a relative stable trend characterized by still high
terrigenous inputs (Bayon et al., 2012; Figure 3) and low BiSiO$_2$ and TOC values (Schneider
et al., 1997), total dinocyst concentrations reach their maximum. To understand this atypical
total dinocyst concentration signal, indexes of specific diversity and dominance have been
calculated so as to help us discussing periods possibly characterized by cyst advection
(positive correlation between dominance and diversity) and in situ dinoflagellate productivity
(negative correlation between dominance and diversity). Here, signals remain roughly anti-
correlated all over the core, except from 90 cm (Figure 3), perhaps involving massive
advection of *O. centrocarpum* at that time (Figure 3).

***4.2 Dinocyst assemblages***
Based on variations in cyst concentrations and in relative abundances of major species, five
palynozones (A, B, C, D, E; Figure 3) have been established, then subdivided into several
sub-palynozones (1, 2, 3; Figures 4 and 5) thanks to the rest of the assemblage (minor species
always observed with at least >2%; Figures 4 and 5).
Temporal successions between dinocyst species can be observed all along the core. This is
especially obvious regarding a dinocyst group mainly controlled by sea-surface salinity
(Marret and Zonneveld, 2003), including *Spiniferites ramosus*, *Nematosphaeropsis*
*labyrinthus*, *L. machaerophorum*, *O. centrocarpum* and *Operculodinium israelianum* (Figures
3, 4), as well as *Echinidinium* spp. (Figure 5). The first important succession occurred at 37.5
ka BP (limit between palynozones E and D), with a significant drop of the maximal





abundances of the couple *S. ramosus - N. labyrinthus*, then followed by maximal abundances
of *L. machaerophorum* (Figure 4). At 32 ka BP (limit between palynozones D and C), a
second major transition is related to a strong decline of *L. machaerophorum* synchronously
with maximal abundances of *O. centrocarpum* percentages, and then accompanied by *O.*
*israelianum* across whole palynozone C (Figure 4). A third succession (limit between
palynozone C and B) is then operated between *O. centrocarpum* and *Echinidinium* spp.
(Figures 4 and 5) at 15.5 ka cal. BP, while the last major transition (limit B-A) evidences the
important decline of *Brigantedinium* spp. and strong re-increase of *O. centrocarpum* from 6
ka BP onwards, together with the significant occurrence of *Spiniferites pachydermus* near the
start of palynozone A (a2, Figure 4).
Among non-heterotrophics, a group of thermophile species can be described with
*Impagidinium aculeatum, Impagidinium patulum, Spiniferites bentorii*, *Tuberculodinium*
*vancampoae, Spiniferites membranaceus* and *S. pachydermus*. This group also shows obvious
temporal successions across the different palynozones (cf. Figure 4). This is especially true
for *S. pachydermus* at the start of palynozone A (Figure 4). Concerning *Operculodinium*
*aguinawense* (Figure 4), the southernmost occurrences ever recorded of this species is here
observed with marine core KZAI-01. This species only occurs today off the coasts of
Cameroon and eastern Nigeria, in a small area encompassing GeoB4905 core (Marret and
Kim, 2009). Over the last 15.5 ka, variations of *O. aguinawense* percentages are relatively
well correlated with *Spiniferites mirabilis* ones, especially across the Last Deglaciation.
Today, both species are restricted to the same area along the north equatorial African coast
(Zonneveld et al., 2013).
Among heterotrophics, coastal taxa such as *P. schwartzii, Selenopemphix nephroides*, and
especially *Xandarodinium xanthum* as well as *Quinquecuspis concreta* (Figure 5), show
extremely close occurrences all along the core. This is especially obvious between 37 and 7



ka BP (Figure 5), and maximal abundances of these species are recorded around 36-32 ka, 25-
20 ka and 15.5-7 ka BP (Figure 5). Also, another important feature is the disappearance of *P.*
*schwartzii* around 35 ka BP, synchronously with *S. nephroides* significant increase at that time
(Figure 5; limit between sub-palynozones d2 and d1).

**5. DISCUSSION**
***5.1. Orbital control on past dinoflagellate productivity regimes***
*5.1.1. Dinoflagellate productivity on the Congolese margin: Congo River versus upwelling*
*dynamics*
Over the last glacial cycle, it is commonly admitted that higher primary productivity
conditions in the intertropical band occurred during periods of global cooling such as the
LGM or Greenland Stadials (GS, including Heinrich Stadials or HS), in response to
intensified upwelling cells. Inversely, during warmer and wetter periods such as Greenland
Interstadials (GI) or the Holocene characterized by higher riverine inputs (Dupont et al., 1998;
Shi et al., 1998; Dupont and Behling, 2006; Kim et al., 2010; Zonneveld et al., 2013), primary
productivity is low.
Within our dinocyst record, higher dinoflagellate productivity seems to be recorded during the
last glacial until 15.5 ka BP (high cyst concentrations and), consistently with high values of
Biogenic Silica (BiSiO$_2$) and Total Organic Carbon (TOC) observed in a neighbour core
(GeoB 1008; (Schneider et al., 1997). Furthermore, *Trinovantedinium applanatum*, a typical
well-known coastal upwelling species (Marret and Zonneveld, 2003), mainly occurred
between around 28 and 19 ka BP (palynozones c3 to c2; Figures 5 and 6), consistently with
the idea of stronger upwelling cells across glacial maxima, and more specifically here during
MIS 2, in a dry context characterized by weak terrigenous supplies to the Congo margin, and
cold sea-surface conditions as evidenced through *O. centrocarpum* higher percentages.





At the onset of the Last Deglaciation, around 15.5 ka BP, a quasi-disappearance of the *T.*
*applanatum* signal is observed (Figures 5 and 6). However, heterotrophic percentages remain
high and are even characterized, between 15.5 and 7 ka BP (sub-palynozones b2 and b3), by
the highest values ever recorded (Figures 5 and 6). This brings us to consider, at that time,
another major source of nutrients to the ocean than upwelling cells. The relatively good
consistency between major element terrestrial signals (cf. XRF ratios in KZAI-01 core, Figure
6), heterotrophic (*Brigantedinium* spp.) as well as fluvial-sensitive cyst (*Echinidinium* spp.,
river-plume taxa) percentages, suggests that nutrient-rich freshwater discharges from the
Congo River probably acted as a major driving factor for promoting dinoflagellate
productivity in the study area, especially across the Last Deglaciation, but also during MIS 3
(Figures 5 and 6). Furthermore, between 15.5 and 7 ka BP, continental shelf reworking,
induced by the post-glacial sea-level rise, may have also represented an additional source of
nutrients to the ocean (Marret et al., 2008), then also contributing to slightly enhanced
dinoflagellate productivity at that time (Figures 5 and 6).

*5.1.2 Precession versus Obliquity accounting for different fluvial regimes*
The influence of orbital forcing in low latitude atmospheric processes is still a matter of
debate. The tropical response to obliquity forcing appears to be the remote influence of high-
latitude glacial ice-sheet oscillations (deMenocal et al., 1993), in parallel with significant
changes in cross-equatorial insolation gradient (Bosmans et al., 2015). Precession forcing is
more important in low latitude moisture changes, i.e warmer and wetter conditions in the
hemisphere where summer solstice occurred at the Earth perihelion (Merlis et al., 2012).
Furthermore, it was evidenced that the combination precession/obliquity has also a great
influence in the monsoon oscillations with a significant prevalence of the precession forcing
(Tuenter et al., 2003). More precisely, minima of precession would correspond to an



intensification of the monsoonal activity, and obliquity would tend to mitigate (minima of
obliquity) or enhance (maxima of obliquity) the initial precession forcing (Tuenter et al.,
2003). The orbital variations have therefore changed significantly the latitudinal widespread
of precipitations in consequence of oceanic heat gradient variations (Stager et al., 2011;
McGee et al., 2014).
In our dinocyst record, significant occurrences of fluvial-sensitive cysts (especially *L.*
*machaerophorum* and *Echinidinium* species) appear to correspond to minima of precession,
thus suggesting wetter conditions in the study area (Figure 6). This is especially observed
during the last Deglaciation-early Holocene between 16 and 6 ka BP (with the prevalence of
*Echinidinium* spp.), as well as during the MIS 3 interval between 39 and 32 ka BP (with the
prevalence of *L. machaerophorum*). Superimposed on this general scheme, a combination
"minimum of precession-maximum of obliquity" would explain the optimal orbital
combination for high moisture conditions according to Tuenter et al. (2003). This
configuration indeed occurred between 16 and 6 ka BP in our dinocyst results and
corresponds to the maximal recorded values of fluvial-sensitive cysts (*Echinidinium* spp.) in
combination with the highest values of heterotrophic cyst percentages (mainly including
*Brigantedinium* spp.; Figure 6).
The minimum of precession recorded during MIS 3 (Figure 6) is characterized by a
decreasing trend of Earth's obliquity, and is also consistently characterized by a weaker Ti/Ca
ratio and associated lower surface productivities between 39 and 32 ka BP (Figure 6). Despite
the austral location of KZAI-01, dinocyst assemblages indicate wetter conditions during
precession minima (Figure 6), i.e when Earth perihelion occurred during northern summer
solstice, with consequently associated drier conditions in the Southern Hemisphere (Merlis et
al., 2012). Conversely, maxima of precession, supposed to be favourable for wetter conditions
in the Southern Hemisphere, correspond to periods with lower terrigenous inputs, and



especially between 44 and 39 ka BP (subpalynnozone e2) and between 25 and 16 ka BP
(subpalynozones c2 and c1; Figure 6).

*5.1.3. The atypical signature of the MIS 2*
In the tropics, during MIS2, the latitudinal contraction of the TR resulted in colder conditions
on the continent (Powers et al., 2005; Tierney et al., 2011; Loomis et al., 2012) with the
establishment of open landscape (Anhuf et al., 2006), and cold surface waters (deMenocal et
al., 2000; Syee Weldeab, 2005; Shakun and Carlson, 2010). This general and commonly
admitted pattern is in agreement with the low terrigenous signal recorded on core KZAI-01
(Figure 6) that suggests reduced weathering conditions combined with lower terrestrial
erosion at that time. Paleo precipitation reconstructions (Bonnefille and Chalié, 2000) also
suggest generally low mean values of precipitations in the Congo Basin, however
characterized by a complex pattern oscillating between slightly wetter and drier conditions. In
our dinocyst record, slight occurrences of *T. applanatum*, *Selenopemphix quanta* (Figure 5)
and cysts of *Pentarpharsodinium dalei* (Annexe 1) are consistent with the tropical
climateuring glacial period, mainly influenced by upwelling mechanisms under dry climate.
However, between 25 and 17 ka BP (sub-palynozones c2 and c3; Figure 6), low abundances
of *Echinidinium* species as well as high percentages of *L. machaerophorum* (up to 50 %)
would suggest strengthened river discharges and thus wetter conditions consistently with the
general pattern of Austral moisture during maximal values of the precession index. The
lowered sea level influence on dinocyst assemblages, at that period, cannot be however totally
excluded regarding the neritic ecology of *L. machaerophorum*. However, another atypical
dinocyst signature of MIS2 relies on the occurrence of thermophile species (*S. mirabilis*, *S.*
*membranaceus*, *S. bentorii* and *T. vancampoae*; Figures 4 and 5) which mainly occurred
between 21 and 17 ka BP, after a gradual increase noted from the beginning of the LGM





(Figures 4 and 5). The southward shift of the TR and the equatorial warm waters until 2°S
(Arbuszewsky et al., 2013) may have brought heat and moisture within the study area while
other parts of the Equatorial Atlantic remained colder and drier (Stager et al., 2011). This
pattern can possibly be explained by the cross-equatorial location of the Congo Basin, also
benefiting from southern hemisphere wetter configurations. Northern Congo Basin
corresponds to a tierce of the whole surface but northern tributary rivers contribute to an half
of the total discharge (Bultot, 1971; Lempicka, 1971), it is however important to underline the
greater influence of northern rivers in comparison with austral ones within the Congo Basin;
terrigenous inputs to the ocean will then be more important when the northern basin will be
fed by strengthened precipitations in a boreal context of precession minima.

*5.1.4 Eastern and Western African monsoons: complex interferences in the Congo Basin*
The large scale of the Congo Basin raises the question of the complex interferences between
Western and Eastern African monsoon systems, i.e. the atmosphere above the catchment area
is divived by the Congo Air Boundary (CAB) convergence zone (Tierney et al., 2011),
displaying the border between the western and eastern African monsoon. Past oscillations of
these different monsoon clusters have been simulated (Caley et al., 2011; Figure 6) through
paleo river discharges of the Niger (Western African monsoon) and of the Nile (Eastern
African monsoon). As mentioned above, dinocyst river-plume assemblages of core KZAI-01
develop strongly in response to boreal summer river discharges linked with precession
minima (Figure 6), suggesting that the Western African monsoon can be considered as the
main forcing for northern summer rainfalls in the Congo Basin. This common pattern is
particularly well highlighted during the Last Deglaciation when river-plume taxa abundances
increase in parallel with terrigenous signals shortly after the increase of the Western African
monsoon around 16 ka BP (Figure 6). Furthermore, the maximum of the West African





monsoon activity, that occurred between 8 and 6 ka BP (Figure 6), also corresponds with the
highest occurrences of *O. aguinawense*, evidencing a great relationship between the western
African monsoonal forcing and the establishment of near equatorial conditions during this
period (Marret and Kim, 2009).
However, the suitable relationship described above between Western African monsoon signal
and dinocyst assemblages is less evident during the recorded wetter interval ranging from 39
to 27.5 ka BP (Figure 6). Our dinocyst data would indeed suggest a better correlation with the
maximum of the Eastern African monsoon signal (Figure 6) while the Western one remained
weakened. This pattern is well correlated with pollen-inferred paleo precipitations data
extracted from Burundi mounts (Bonnefille and Chalié, 2000), which display higher
precipitations during this interval, also in accordance with strengthened Eastern African
monsoons (Figure 6).

***5.2. Sub-orbital variations over the last 20 ka***
*5.2.1. The Last deglaciation*

*The tropical response of Heinrich Stadial 1 (HS1)*
Between 18 and 15.5 ka BP, thermophile and river-plume species abundances sharply dropped
while *O. centrocarpum* reached very high percentages (up to 50%) at that time (sub-
palynozone c1; Figure 7). Combined with low occurrences of *T. applanatum* (Figure 7), *O.*
*centrocarpum* here suggest significant SST cooling, probably induced by an intensification of
the BC activity in the area, and associated with enhanced upwelling cells activity. This
dinocyst pattern is consistent with previous observations that described a strong drought on
the African continent (Stager et al., 2002; Stager et al., 2011; Bouimetarhan et al., 2012;
Weldeab et al, 2012), as recorded through low precipitations (Bonnefille and Chalié, 2000;



Schefuss et al., 2005; Figure 6) associated with a continental and marine cooling ranging
between 1 and 2°C below LGM mean values (Mueller et al., 1998; Weldeab et al., 2007;
Powers et al., 2008; Weldeab et al., 2011; Shannon et al., 2012).
This cool and dry event appears synchronous with a massive advection of freshwater melting
that occurred in the North Atlantic Basin during Heinrich Stadial (HS) 1. The tropical
response of HS 1 would then consist in the southward shift of the TR (Arbuszewski et al.,
2013; McGee et al., 2014), involving a contraction of the latitudinal belts (Stager et al., 2011)
and weakened monsoons during this period. It is interesting to note that, while dinocysts
evidence a marked sea-surface cooling, isotopic signals from nearby core GeoB6518-1
(Schefuss et al., 2005; Figure 7) suggest a steady increase in tropical moisture all along HS1.
This implies a fundamental divergence between marine and continental compartments across
the Last Deglaciation.

*The equatorial signal of increasing deglacial warming at 15.5 ka BP*
Around 15.5-15 ka BP, the equatorial deglacial transition occurred in parallel with a global
warming (Syee Weldeab, 2005; Weijers et al., 2007; Leduc et al., 2010), linked with the
Northern Hemisphere July insolation increase. This resulted in a northward shift of the TR
(Arbuszewski et al., 2013; McGee et al., 2014) and thus strengthened monsoon activities.
Our dinocyst data also show a significant transition at around 15.5 ka BP (limit between
palynozones B and C; Figure 7) with the rapid increase of *Brigantedinium* spp. and
*Echinidinium* spp. percentages (Figure 7). Their modern distributions in the tropics are both
related to nutrient-enriched waters and, more specifically for *Echinidinium* spp., to high river-
discharges (Zonneveld et al., 2013). This is consistent with the strong increase of terrigenous
inputs observed at that time in the same study core (Bayon et al., 2012; Figure 7). The
equatorial species *O. aguinawense* also occurred shortly at around 15.5 ka BP (Figure 7),





suggesting a short high near-equatorial moisture event. *L. machaerophorum* abundances also
re-increased at 15.5 ka BP but remained low in comparison with glacial ones, suggesting a
specific switch in fluvial-sensitive dinocyst tracers between *L. machaerophorum* (glacial) and
*Echinidinium* spp.(across and after the Last Deglaciation).
Among the thermophile species, *Selenopemphix nephroides* and especially *Stelladinium reidii*
are the most obvious signals of the post 15.5 ka BP deglacial warming (Figure 7). Both
species are also considered as good tracers for high regimes of trophic conditions (Zonneveld
et al., 2013), in agreement with the recorded surface nutrient enrichment previously suggested
during this period (cf. *Brigantedinium* spp. and *Echinidinium* spp.). .

*The tropical response of the Younger Dryas (YD)*
Significant dinocyst changes occurred between around 13 and 11.5 ka BP in both dinocyst and
geochemical records (Figure 7). The significant drop of XRF Ti/Ca ratio evidences a reducing
of terrigenous inputs, while percentages of thermophile species *S. mirabilis*, *S. nephroides* and
*S. reidii* strongly decrease, then suggesting a significant cooling of surface waters in the study
area. The recorded cooling would be in agreement with the Younger Dryas (John Lowe and
Hoek, 2001) timing (Figure 7). However, high abundances of *Echinidinium* spp.and
*Brigantedinium* spp. during this interval suggest that nutrient-enriched river discharges still
occurred at that time (Figure 7). Our recorded tropical wetter conditions could be explained
by a suitable location of the TR above the Congo Basin, between Holocene and LGM mean
location (Arbuszewski et al., 2013; McGee et al., 2014). Furthermore, the weakening of the
deglacial sea-level rise during this period (Grant et al., 2012) and therefore the decrease of
associated continental shelf reworking (Marret et al., 2008) could explain the observed drop in
terrigenous inputs and the long-term decreasing trend of *Echinidinium* spp. (Figure 7). The
absence of *T. applanatum* during the tropical response of the YD would also suggest the





absence of upwelling cells in the study area (Figure 6). Nevertheless, high abundance of *S.*
*quanta* and *S. membranaceus* (Figures 5 and 7), generally well abundant in the vicinity of
seasonal upwelling cells (Marret and Zonneveld, 2003), may suggest the development of
seasonal coastal upwelling close to the study area, probably related to the suborbital-scale
northward shift of the ABF (Jansen et al., 1996).

*5.2.2 The Holocene*
The weak chrono-stratigraphic constraint of the Holocene (cf. Figure 2) leads to take great
caution in the interpretation of detailed specific events. However, some major subdivisions
(Early-, Mid-, and Late-Holocene) can be generally discussed (Figure 7).

*The Early Holocene and African Humid Period*
Across the Holocene, the African Humid Period (AHP) is a significantly warmer and wetter
period that occurred between around 14.5 and 5 ka BP (deMenocal et al., 2000; Shanahan et
al., 2015). At that time, the TR was characterized by a wider latitudinal extension up to
several degrees poleward (Stager et al., 2011; Arbuszewski et al., 2013; McGee et al., 2014).
Previous dinocyst studies, showed that the AHP was characterized by the gradual bloom of
thermophile (*S. mirabilis*) and low-salinity (*O. aguinawense*) species, induced by
strengthened river discharges from the beginning of the Holocene (Dupont and Behling, 2006;
Kim et al., 2010; Marret et al., 2013).
Similarly to these published data, our record also evidences a strengthening of nutrient-
enriched river discharges from the onset of the last deglaciation (Figure 7; cf. subchapter
5.2.1). However significant occurrences of *O. aguinawense* between around 11 and 2 ka BP
(Figure 7) delimit the effective wettest period also characterized by the highest abundances of
both mesotrophic and eutrophic thermophile species (Figure 7). High SST recorded at the



beginning of the Holocene are also well correlated with alkenone SST reconstructions from
core GeoB6518 (Schefuß et al., 2005) synchronously with the Early Holocene timing (Figure
7). It is also interesting to note that, during this Holocene climatic optimum, our dinocyst data
show a sharp drop of *Echinidinium* abundance between 8 and 7 ka BP, synchronously with a
drop of thermophile species (Figure 7). This could suggest a thousand years-long cooler and
drier event that occurred during the Early and Mid-Holocene transition (Walker et al., 2012).

*The Mid-Holocene transition and the end of African Humid Period*
The timing of the AHP termination significantly changes according to authors and study sites
(Figure 7), i.e around 2.5 ka BP (Kröpelin et al., 2008; Lézine et al., 2013; Shanahan et al.,
2012; Lebamba et al., 2012), 4 ka BP (deMenocal et al., 2000; Hély et al., 2009; Tierney and
deMenocal, 2013; Shanahan et al., 2015), or even earlier around 5.3 ka BP (Lézine et al.,
2005). The length of the AHP termination also changes significantly, from a few centuries to a
few thousand years (Figure 7) according to the references mentioned above.
In our data, we observe two abrupt degradation steps during a millennial-scale heat and
moisture mitigation. The first mitigation occurred abruptly between 7 and 6 ka BP (transition
between sub-palynozones b1 and b2), illustrated by the sharp drop of heterotrophic taxa
percentages, especially *Brigantedinium* species, *S. reidii* and *S. nephroides* (Figure 7), in
parallel with high abundances of *O. centrocarpum*. This suggests an environmental change
from eutrophic to lower nutrient-enriched surface waters, probably allowing the observed
development of mesotrophic taxa, such as *S. mirabilis* (Figure 7). This mitigation does not
appear as a dry event, due to: i) the persistence of high *Echinidinium* spp. abundances, with
however a long-term decreasing trend obvious since 15.5 ka BP (Figure 7) and ii) the
persistence of *O. aguinawense* (today related to near-equatorial hydrological conditions)
which reached its highest abundances during this interval (sub-palynozone b1, Figure 7).



The second mitigation occurred abruptly between 4 and 3.5 ka BP (transition between
palynozones A and B; Figure 7), as displayed by a general drop of both heterotrophic and
thermophile cyst percentages, while *O. centrocarpum* rapidly became the major dinocyst
species (Figure 7). As we discussed above from crossed information related to total dinocyst
concentrations and community indexes (dominance *versus* diversity; cf. Section 4.1 and
Figure 3), the interval ranging from 4 to 2.5 ka BP (sub-palynozone a2; Figure 7) is probably
characterized by massive advection of *O. centrocarpum* cysts. However, removing *O.*
*centrocarpum* from abundance calculations of other taxa will not remove the observed shift
discussed above for heterotrophic and thermophile dinocysts and clearly related the 4-3.5 ka
BP period.

*The Late Holocene*
Right after 3.5 ka BP the interval appears to be one of the coolest and driest periods ever
recorded in core KZAI-01, as it was observed elsewhere in tropical Atlantic latitudes (Marret
et al., 2006). This could be the consequence of a strengthened advection of the BC northward,
maybe also related to a 4° northward shift of the ABF, well recorded during the Mid to Late
Holocene (Jansen et al., 1996).

Finally, since 2.5 ka BP (subpalynozone a1), a recovery of *L. machaerophorum* and
*Echinidinium* spp.percentages is observed in parallel with low occurrences of *O. aguinawense*
(Figures 5 and 7), suggesting a slight re-increase of wetter conditions. However, despite the
general warming observed in several SST reconstructions in tropical studies over this period
(Schefuß et al., 2005; Weldeab et al., 2005; Dyez et al., 2014), all thermophile cyst
percentages remain zero or very low (Figure 7). The recovery of wetter conditions may be
explained by the optimum of the precession index reached around 3 ka BP (Figure 6), which



implies the correspondence between austral summer and Earth perihelion and allows the
establishment of wetter and warmer conditions in the Southern Hemisphere.
The Holocene as recorded in core KZAI-01 can be then divided into three major periods. The
earliest interval (11-6.5 ka BP) is also the warmest and wettest period, followed between 6.5
and 4 ka BP by a mitigated warm and humid period characterized by the progressive recovery
of the BC advection. Finally, after 4 ka BP, a major long-term cooling and drying period is
gradually pondered by a progressive recovery of river discharges in the study area since 2.5
ka BP.

**6. CONCLUSION**
Dinocyst assemblage analysis conducted on core KZAI-01 has permitted an investigation of
land-sea-atmosphere linkages off the Congo River mouth over the last 44 ka. Our dinocyst
data evidence a great influence of nutrient-rich river discharges induced by the Tropical
Rainbelt latitudinal migrations, themselves forced by different orbital configurations, and
especially regarding the combination "precession minima - obliquity maxima". Furthermore,
while most of tropical studies describe the LGM as a "cold and dry" period in the tropics,
dinocyst assemblages evidence here a slightly warmer and wetter period than it was expected.
The LGM appears to be a complex period characterized by a southward latitudinal shift of the
monsoonal belt and of warm surface waters, bringing heat and moisture. This illustrates a
complex scheme that would deserve model simulations and unravelling precise underlying
mechanisms and impacts that occurred across this specific climate interval.
This high-resolution study has also permitted to discriminate major climatic periods of the
Last Deglaciation in good correspondence with Northern Hemisphere high latitude millennial-
scale oscillations. We also discussed the timing of the equatorial response of the African
Humid Period and the two-steps mitigation of heat and moisture conditions in the study area.



Further work will involve a regional-scale study including other dinocyst records to
reconstruct sea-surface environments in relationship with Tropical Rainbelt latitudinal shifts
along the African Western façade, as well as model-data inter-comparisons for different
snapshots across the last glacial, deglacial and Holocene periods.

**7. ACKNOWLEDGEMENTS**
W. Hardy's PhD was funded by Brittany Region and this work was supported by the
"Laboratoire d'Excellence" LabexMER (ANR-10-LABX-19) and co-funded by a grant from
the French government under the program "Investissements d'Avenir". We thank Mr. B.
Dennielou (Ifremer, Brest) for access to core KZAI-01.






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



## 9. TABLE AND FIGURE CAPTIONS


### 9.1. Table caption


**Table 1** : Inventory of all dates obtained in the core KZAI-01 : 14C AMS datations obtained
from carbonate materials (Bayon et al., 2012), 14C AMS datations obtained from bulk organic
matter (Bayon et al., 2012) and finally dates obtained from tuning with core GeoB6518-1
(Schefuß et al., 2005; Bayon et al., 2012). Rejected dates are displayed in red.

### 9.2. Figure captions


**Figure 1**: Map showing locations of KZAI-01 core and other cores mentioned in the text:
GeoB6518 (Schefuß et al., 2005; Bayon et al., 2012), GeoB1008 (Schneider et al., 1997) and
GITANGA2 (Bonnefille and Chalié, 2000). The general pattern of present-day surface ocean
currents of the adjacent Atlantic Ocean is extracted from Lass and Mohrholz (2008) and
includes: the Guinea Current (GC), the northern (nSEC), equatorial (eSEC), central (cSEC),
and southern (sSEC) South Equatorial Current, the Angola Current (AC), the Angola-
Benguela Front (ABF), the Benguela Current (BC) and the Agulhas Current (AgC). Orange
lines indicate warm currents and blue lines cold currents. Green zones correspond to
upwelling zone (BUS : Benguela upwelling system) and oceanic domes (AD : Angola Dome,
ED : Equatorial Dome; Voituriez, 1981; Lass and Mohrholz, 2008). Black dashed line display
the mean location of the ITCZ during July and January (Collier and Hughes, 2011). Red 5°C-
interval isolines correspond to annual mean SST (Hirahara et al., 2013). Vegetation covering
(in % per surface unity) is extracted from (Hansen et al., 2013) dataset.

**Figure 2**: Age model established on linear regression calculated from AMS $^{14}$C datations on
carbonate (red squares; cf. Table 1). Blue squares correspond to $^{14}$C datations extracted from
organic matter (Bayon et al., 2012), not taken into account for the age model. Green squares
correspond to dates obtained by tuning with core GeoB6518, on the basis on similar trend
observed in Ti / Ca XRF ratios extracted from respective cores. Grey band corresponds to the
range error of calibrated dates, and purple lines correspond to the sedimentation rates (cm/ka).


**Figure 3**: Comparisons between total dinocyst concentration in the sediment (cysts / cm$^3$), the
proportion of non-heterotrophic taxa concentration in the whole assemblage and the dominant
species carrying this concentration, i.e *Lingulodinium machaerophorum and Operculodinium*
*centrocarpum*. The same approach is applied for heterotrophic taxa, with the comparison
between total heterotrophic concentrations in the sediment in the view of heterotrophic
dominant species concentration, i.e *Brigantedinium* spp.and *Echinidinium* spp. The cited
dominant species abundances are illustrated in cumulated percentages. To discuss the
relationship between primary productivity, dinocysts concentrations and terrigenous dilution,
the Ti/Ca XRF ratio of the core KZAI-01 is displayed, in addition with biogenic silica and
total organic matter signals extracted from core GeoB1008 (Schneider et al., 1997). We added
the specific diversity and dominance index to discuss the potential advection of allogeneic
dinocysts in the study. Red dashed lines correspond to major transitions in total dinocysts
concentration in the view of known major environmental shifts. Major palynozones (ABCDE)
boundaries are established on the basis of major dinocysts concentration transition periods.



**Figure 4**: Detailed non-heterotrophic major species abundances in view of total dinocysts in the sediment (cysts/cm$^3$). Some species have been grouped, such as *Spiniferites ramosus* and *Spiniferites bulloides*, grouped into *Spiniferites ramosus*, and *Nematosphaeropsis labyrinthus* grouped with *Nematosphaeropsis lemniscata*. Palynozones (A to E) have been established according to the major dinocyst variations in absolute concentrations and relative abundances, with minor subdivisions (Ax-Ex). Species are displayed here and classified according to observed temporal successions, underlined by black arrows. Black arrows represent temporal successions between species abundances.

**Figure 5**: Detailed heterotrophic taxa abundances in parallel with abundances of total heterotrophic taxa and heterotrophic concentrations in cysts/cm$^3$. Some species have been grouped, such as: *Echinidinium* spp.(*E. aculeatum, E. delicatum, E. granulatum* and *E. transparentum*). *Lingulodinium machaerophorum* is displayed here with *Echinidinium* spp.regarding their river-plume affinity. Palynozones (A to E) have been established according to the major dinocyst variations in absolute concentrations and relative abundances, with minor subdivisions (Ax-Ex). Heterotrophic species are displayed here and classified according to observed temporal successions, underlined by black arrows. Black arrows represent temporal successions between species abundances.

**Figure 6**: Comparison between total heterotrophic abundance and upwelling activity displayed by *Trinovantedinium applanatum* (Marret & Zonneveld, 2003). Congo River discharges are displayed on KZAI-01 core by river-plume sensitive species *Echinidinium* spp. and *Lingulodinium machaerophorum,* Coenobia of *Pediastrum* and terrigenous inputs (Ti/Ca XRF ratio, quantitative measurements of major elements Al/K and Al/Si; Bayon et al., 2012). Relations between river discharges and paleomonsoon activity are displayed through rainfall anomalies in Burundi mounts (Bonnefille & Chalié, 2000: the threshold with positive anomalies in green and negative anomalies in orange is calculated from mean glacial values) and regional-scale monsoon reconstructions (Western and Eastern African Monsoon; Caley et al., 2011: maximal monsoon regimes are underlined in green). Orbital parameters such as Obliquity and Precession (Berger and Loutre, 1991) are also displayed with precession minima highlighted in green, and obliquity maxima highlighted in green. Green bands correspond to major orbital-scale moisture conditions. dD on Alkane C29 from core GeoB6518 is displayed in red with pollen-inferred paleo precipitation reconstructions.

**Figure 7:** Temporal focus on the last 20 ka BP. Sea-surface Temperature changes are discussed with major dinocyst species, classified according their trophic affinity: *Spiniferites mirabilis*, *Spiniferites membranaceus*, *Selenopemphix nephroides* and *Stelladinium reidii*. Sea-surface salinities changes are discussed with: sum of Echinidinium species, Operculodinium aguinawense, Lingulodinium machaerophorum, in addition with stable isotopic signal from core GeoB6518-1 (Schefuss et al., 2005). Upwelling activity and Benguela advection are discussed with *Trinovantedinium applanatum* and *Operculodinium centrocarpum* respectively, and marine food abundance with *Brigantedinium* spp. Ti/Ca XRF ratio of core KZAI-01 represents past terrigenous supplies. d18O GICC05 is displayed here (Svensson et al., 2008) with the Deglaciation-Holocene subdivisions (Walker et al., 2012) : Last Glacial Maximum (LGM), Heinrich Stadial 1 (HS1), Bølling-Allerød (B/A), Younger Dryas (YD), Early Holocene (EH), Mid-Holocene (MH) and Late Holocene (LH). Blue bands correspond to cold and dry events recorded with dinocyst assemblages. African Humid Period (AHP) terminations are also depicted in the figure according to literature (orange bars) :



1°(Kröpelin et al., 2008), 2° (Shanahan et al., 2015), 3° (Hély et al., 2009), 4°(Lézine et al.,
2005), 5°(Lebamba et al., 2012), 6°(Lézine et al., 2013), 7°(Tierney and deMenocal, 2013),
8°(Shanahan et al., 2012), 9° (deMenocal et al., 2000)
Established palynozones subdivisions are also displayed (aX, bX; cf. Figures 4 and 5).






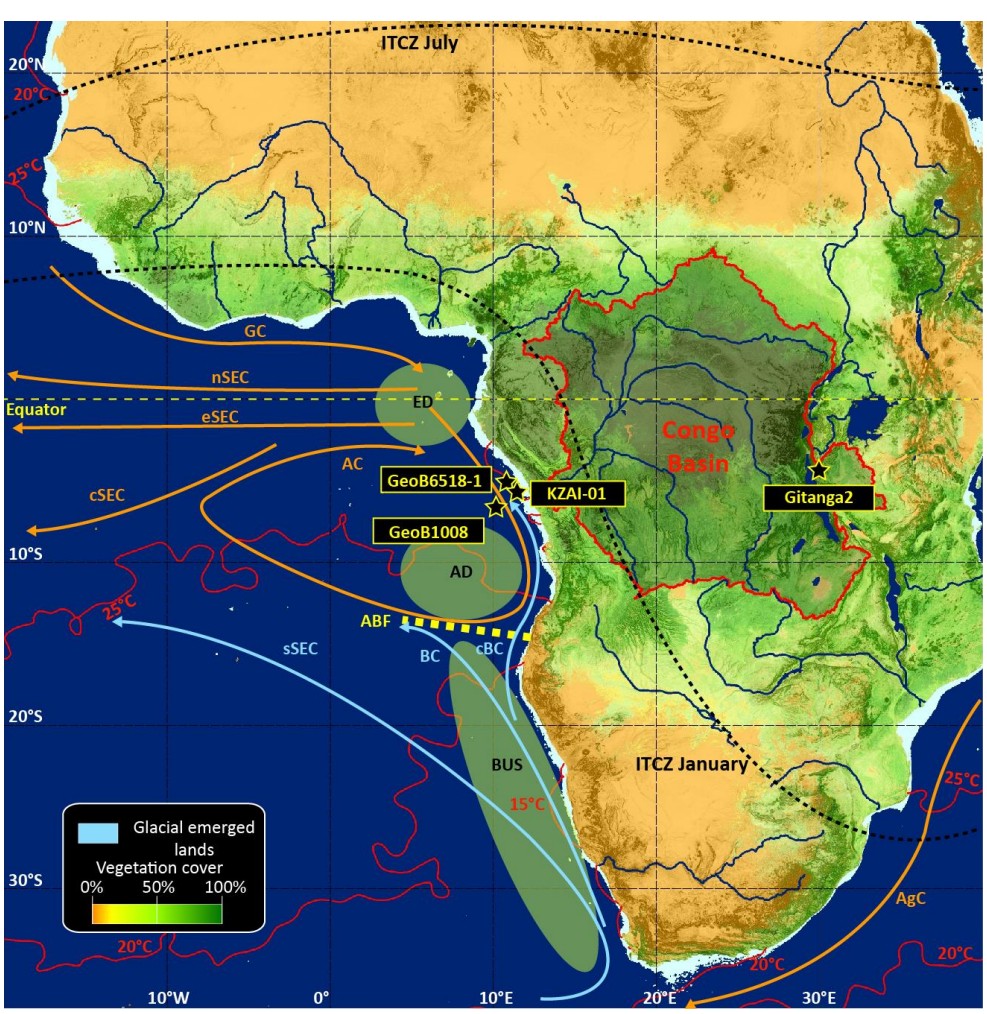

Figure 1




Figure 2





Figure 3





Figure 4





Figure 5








Figure 6





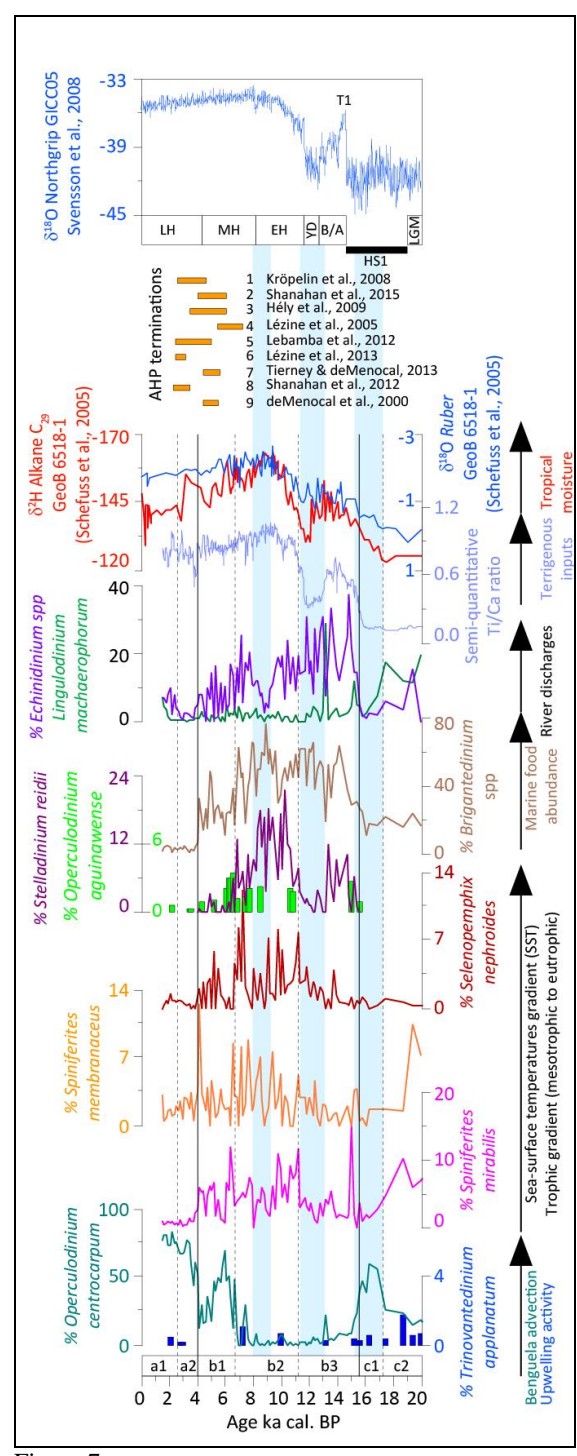

Figure 7



| Depth | Material | Laboratory number | Calendar age (BP) | Calibrated $^{14}$C (cal. BP) | Data | Comments |
|---|---|---|---|---|---|---|
| 10 | Mixed marine carbonate | UtC-9311 | 2172 +/- 39 | 1763 | Bayon et al., 2012 | |
| *13* | *Bulk organic matter* | *Poz-40293* | *1610 +/- 30* | *1170* | *Bayon et al., 2012* | Not used |
| *18* | *Bulk organic matter* | *Poz-40295* | *2310 +/- 30* | *1921* | *Bayon et al., 2012* | Not used |
| *26* | *Bulk organic matter* | *Poz-40296* | *2545 +/- 30* | *2216* | *Bayon et al., 2012* | Not used |
| *37* | *Bulk organic matter* | *Poz-40297* | *3210 +/- 30* | *3024* | *Bayon et al., 2012* | Not used |
| *51* | *Bulk organic matter* | *Poz-40298* | *3770 +/- 30* | *3713* | *Bayon et al., 2012* | Not used |
| *70* | *Bulk organic matter* | *Poz-40299* | *4435 +/- 35* | *4636* | *Bayon et al., 2012* | Not used |
| *122* | *Bulk organic matter* | *Poz-40300* | *5970 +/- 40* | *6380* | *Bayon et al., 2012* | Not used |
| *190* | *Mixed marine carbonate* | *UtC-9312* | *8710 +/- 60* | *9369* | *Bayon et al., 2012* | Rejected |
| *196* | *Bulk organic matter* | *Poz-40301* | *8080 +/- 40* | *8527* | *Bayon et al., 2012* | Not used |
| 265 | Tuning with GeoB6518 core | | | 9361 | This paper | |
| *269* | *Bulk organic matter* | *Poz-40302* | *9790 +/- 50* | *10727* | *Bayon et al., 2012* | Not used |
| *305* | *Bulk organic matter* | *Poz-40389* | *10400 +/-* | *11503* | *Bayon et al., 2012* | Not used |
| 322 | Tuning with GeoB6518 core | | | 11428 | Bayon et al., 2012 | |
| 356 | Planktonic foraminifera | Poz-20108 | 10930 +/-50 | 12444 | Bayon et al., 2012 | |
| *372* | *Bivalv* | *Poz-73781* | *13450 +/-70* | *15598* | *This paper* | Rejected |
| 444 | Tuning with GeoB6518 core | | | 15756 | Bayon et al., 2012 | |
| 456 | Planktonic foraminifera | Poz-20109 | 13950 +/-70 | 16328 | Bayon et al., 2012 | |
| 522 | *Bolivina spatulatha* | Poz-73782 | 20800 +/- | 24575 | This paper | |
| 585 | Planktonic foraminifera | Poz-20110 | 23020 +/- | 26917 | Bayon et al., 2012 | |
| 622 | Bivalv | Poz-73783 | 24870+/- | 28454 | This | |
| 678 | Mixed marine carbonate | UtC-9314 | 28240+/-280 | 31812 | Bayon et al., 2012 | |
| 851 | Mixed marine carbonate | UtC-9315 | 31800+/-400 | 35350 | Bayon et al., 2012 | |





| 915 | Tuning with GeoB6518 core | | | 38875 | This paper | |
|-----|---------------------------|--|--|-------|------------|--|
| 962 | Tuning with GeoB6518 core | | | 41345 | This paper | |

Table 1