# Peer review of "Dinocyst assemblage constraints on oceanographic and atmospheric processes in the Eastern Equatorial Atlantic over the last 44 ky"

_Biogeosciences, 2016_

## Referee Comment (RC1) · Anonymous Referee #1 · 9 Jun 2016

Please see attached documents

Please also note the supplement to this comment:
http://www.biogeosciences-discuss.net/bg-2016-148/bg-2016-148-RC1-supplement.pdf
* * *
[Figure]

Evaluation report on manuscript BG-2016-148 "**Dinocyst assemblage constraints on oceanographic and atmospheric processes in the Eastern Equatorial Atlantic over the last 44 ka**"

**General appreciation**

This manuscript presents evidence of orbital forcings on oceanographic changes recorded by dinocyst assemblages in a sediment core collected on the Congolese margin. The document presents new and important data on the linkages between Earth's orbital parameters (obliquity and precession) and the evolution of atmospheric continental and sea surface conditions at millennial timescale, and should definitely be published.

The manuscript is well written and was pleasant to read. There are a few typos and syntax error, but other than that it is written in relatively good English. It is clear, concise and straight to the point. If follows a logical progression, make use of the most up to date scientific literature on the subject, and uses adequate methodology to produce the data presented. The discussion is clear and well organized, and all the necessary arguments needed to draw the conclusions are presented in a well-organized and logical progression. All the figures are important for the comprehension of the text, but a few of them will need improvement with respect to the choice of colors (see below). All in all, a very good paper.

**Specific comments**

**Figure 2**. The axes of both Ti/Ca diagrams are drafted in pale grey, which is barely visible on the electronic version, much less on the printed copy. They should be changed to black.

**Figure 4**. It is a very colorful figure but some colors are inadequate: pale blue, pinkish, yellow and lime green over a white background are not legible. Please use more contrasted colors.

**Figure 5**. The taxa *Polykrikos schwartzii* is considered here as "hypersaline", despite the fact Marret and Zonneveld (2003, p. 91) illustrate it as ranging between salinities of ~33 and ~36.5, with maximum abundances around 34.5-35. I do not consider this as "hypersaline". Also, the time period when *P. schwartzii* is abundant corresponds with the consistent presence of freshwater algae (figure 6), although in low concentrations. Could the authors elaborate on that?

**Figure 6**: Same comment as for figure 4 regarding the choice of colors.

**Fig. 1.**

[Figure]

1 **Dinocyst assemblage constraints on**
2 **oceanographic and atmospheric processes in the**
3 **Eastern Equatorial Atlantic over the last 44 ka**
4

5 **Hardy William** [a] *, **Penaud Aurélie** [a] *, **Marret Fabienne** [b], **Bayon Germain** [c], **Marsset**
6 **Tania** [c], **Droz Laurence** [a]
7
8 (a) UMR 6538 Domaines Océaniques, IUEM-UBO, F-29280 Plouzané, France.
9 (b) School of Environmental Sciences, University of Liverpool, Liverpool, L69 7ZT, UK
10 (c) IFREMER, UR Géosciences Marines, BP 70-29280 Plouzané, France
11
12
13 *Corresponding author. Tel.: +33-298-498-741; fax: +33-298-498-760
14 E-mail address: william.hardy@univ-brest.fr
15
16

**Fig. 2.**

**Supplement:**

[Figure]

Université du Québec à Rimouski
**Institut des sciences de la mer de Rimouski**

310, allée des Ursulines, C. P. 3300
Rimouski (Québec)  G5L 3A1, CANADA
Téléphone :   418 724-1650
Télécopieur :  418 724-1842
Courriel : ismer@uqar.ca
www.ismer.ca

Rimouski June 9, 2016

Natascha Töpfer
Copernicus Publications
Editorial Support

Dear Mrs Töpfer,

Please find attached my evaluation report and annotated copy of the manuscript BG-2016-148 "**Dinocyst assemblage constraints on oceanographic and atmospheric processes in the Eastern Equatorial Atlantic over the last 44 ka**", by William H, Penaud A, Marret F, Bayon G, Marsset T and Droz L, submitted for publication in BIogeosciences. I recommend the manuscript for publication, pending minor modifications, which are highlighted in the attached documents.

With my Best Regards,

André Rochon

Professor – Marine Geology
ISMER-UQAR
310 allée des Ursulines
Rimouski QC, Canada
G5L 3 A1
Tel. 1-418-723-1986, ext. 1742
Andre_rochon@uqar.ca

---

## Referee Comment (RC2) · Anonymous Referee #2 · 13 Jun 2016

In this study, Hardy and colleagues present a 44 ka time-series of Dinocyst assemblage carried out in in a sediment core from the Congolese margin. The Dinocyst assemblage are used to explore changes in temperature and nutrient availability due to changes in up-welling and riverine runoof. The major objective of this study is to elucidate hydroclimate over the Congo Basin and oceanic conditions in the eastern equatorial Atlantic. I am not an expert of Dinocysts, but I am familiar with the paleoclimate records from and paleoclimate debate about this region. I think that this study is carefully written and contribute to the emerging discussion regarding the hydroclimate response of the Congo Basin. Below I am listing some major points that the author need to carefully address before their manuscript can be considered for publication in

Biogeoscience.

MAJOR POINTS:

I) Age model: I think it is critical that the authors perform a rigorous error analysis of their age model. I recommend age model uncertainty analysis for every data point using their tie-points and Bayesian statistical error analysis using softwares like "Bacon" (Blaauw and Christen, 2011) or "B-Chron" (Parnell et al., 2008). A graph should be presented showing the age model (sediment depth versus age) and the uncertainty of the age model. When discussing the onset or termination of an event, the median age of onset/termination of event and the uncertainty of the median age point should be provided.

II) The authors describe changes in assemblages or species that would suggest an un "atypical" climate changes that includes warm and wet conditions during the LGM and coldest reconstructed temperature of the record during the late Holocene. In indeed, these inferences are in contradiction with several (mostly geochemical) observations. I suggest therefore, a critical discussion about the robustness and weakness of the interpretation of the assemblage changes. Especially the interpretation of the thermophile and low-salinity assemblages need a critical revisit.

III) The authors need also to provide a more nuanced discussion about the mechanisms that control that the Benguela advection over their core location. In Figure 7, it is suggested that the Benguela advection was severely weakened during 4-5 ka and 7-15 ka BP. How does these interpretations compare with paleclimate records from more southern locations relative to that of the cores described in this study.

IV) A significant part of the Congo Basin is located north of the equator and, therefore, it is a part of the West African monsoon system. I highly encourage the authors to graphically compare their time-series of thermophile and low salinity assemblage/species with the highly resolved runoff and SST records from the Gulf of Guinea (Weldeab et al, 2007a-b, Weldeab et al., 2012a, Weldeab et al., 2012b). It is important that

the authors' micro-paleontolagical approach is compared with and tested against the geochemical proxy records from the same region (see above reference)

V) There are several references cited in the main text but no listed in the reference list. It mismatch can avoided by using one of the several citation software and a careful checking.

VI) Wordings: there are several wordings (admitted, thanks to, climateuring, mitigation, tierce) that the authors need to replace with more appropriated words/phrases. Please change: "Weldeab et al" instead "Syee Weldeab et al.. "biogenic opal" instead "Biogenic Silica (BiSiO2)"

Cited reference:

Blaauw, M., Christen, J.A., 2011. Flexible paleoclimate age-depth models using an autoregressive gamma process. Bayesian Analysis 6, 457-474

Parnell, A. C. et al. (2008). A flexible approach to assessing synchroneity of past events using Bayesian reconstructions of sedimentation history. Quaternary Science Reviews, 27(19-20), 1872–1885.

Weldeab, S. (2012), Timing and magnitude of equatorial Atlantic surface warming during the last glacial bipolar oscillation, Climate of the Past 8, 1705-1716.

Weldeab, S. (2012), Bipolar modulation of millennial-scale West African monsoon variability during the last glacial (75,000–25,000 years ago), Quaternary Science Reviews, 40(0), 21-29.

Weldeab, S., D. W. Lea, R. R. Schneider, and N. Andersen (2007), 155,000 years of West African monsoon and ocean thermal evolution, Science, 316(5829), 1303-1307.

Weldeab, S., D. W. Lea, R. R. Schneider, and N. Andersen (2007), Centennial scale climate instabilities in a wet early Holocene West African monsoon, Geophysical Research Letters, 34(24), L24702.

---

## Author Comment (AC1) · 6 Jul 2016

Dear reviewer,

Thanks for your positive comments, please find attached all the fixed figure with better contrasted colours.

Here follow our responses (we added these answers in .pdf file in supplement)

"The axes of both Ti/Ca diagrams are drafted in pale grey, which is barely visible on the electronic version, much less on the printed copy. They should be changed to black."

We checked it and all lines of diagrams are drawn in black. We use Adobe Illustrator

and it does not find any lines in pale grey. This is probably a visual default resulting from the exportation.

"It is a very colorful figure but some colors are inadequate: pale blue, pinkish, yellow and lime green over a white background are not legible. Please use more contrasted colors."

"The taxon cyst of Polykrikos schwartzii is considered here as "hypersaline", despite the fact that Marret and Zonneveld (2003, p. 91) illustrate it as ranging between salinities of ∼33 and ∼36.5, with maximum abundances around 34.5-35. I do not consider this as "hypersaline". Also, the time period when P. schwartzii is abundant corresponds with the consistent presence of freshwater algae (figure 6), although in low concentrations. Could the authors elaborate on that?"

This was indeed a mistake that we fixed.

"Same comment as for figure 4 regarding the choice of colours."

Also fixed.

Sincerely yours

William Hardy and co-authors

Please also note the supplement to this comment:
http://www.biogeosciences-discuss.net/bg-2016-148/bg-2016-148-AC1-supplement.pdf

—————————————

[Figure]

[Figure]

**Fig. 1.**

[Figure]

**Fig. 2.**

**Fig. 3.**

**Supplement:**

Brest, 1st of July 2016

**Dear editor and reviewers,**

**We have corrected the manuscript taking into account the reviewer's suggestions and we have included all modifications in the revised manuscript. Below you will find our detailed responses to each of the reviewer's comments.**

**We therefore hope to have satisfied the reviewers. For the reader convenience, all our responses are in red in the following text, and all the corrections made to the manuscript are also highlighted in red in the revised version.**

**Furthermore, the quality of the English has been improved throughout the manuscript.**

**Sincerely yours,**

**William Hardy and co-authors**

**Responses to Reviewer 1**

*The axes of both Ti/Ca diagrams are drafted in pale grey, which is barely visible on the electronic version, much less on the printed copy. They should be changed to black.*

We checked it and all lines of diagrams are drawn in black. We use Adobe Illustrator and it does not find any lines in pale grey. This is probably a visual default resulting from the exportation.

*It is a very colorful figure but some colors are inadequate: pale blue, pinkish, yellow and lime green over a white background are not legible. Please use more contrasted colors.*

We fixed it.

*The taxon cyst of Polykrikos schwartzii is considered here as "hypersaline", despite the fact that Marret and Zonneveld (2003, p. 91) illustrate it as ranging between salinities of ~33 and ~36.5, with maximum abundances around 34.5-35. I do not consider this as "hypersaline". Also, the time period when P. schwartzii is abundant corresponds with the consistent presence of freshwater algae (figure 6), although in low concentrations. Could the authors elaborate on that?*

This was indeed a mistake that we fixed.

*Same comment as for figure 4 regarding the choice of colours.*

Also fixed.

**Responses to Reviewer 2**

*"Age model: I think it is critical that the authors perform a rigorous error analysis of their age model. I recommend age model uncertainty analysis for every data point using their tie-points and Bayesian statistical error analysis using softwares like "Bacon" (Blaauw and Christen, 2011) or "B-Chron" (Parnell et al., 2008)"*

We have plotted the $2\sigma$ range error in grayscale envelop around mean calibrated ages. The grey band plotted on the Age - Depth figure (first version) only corresponds to the $^{14}$C error.

*"When discussing the onset or termination of an event, the median age of onset/termination of event and the uncertainty of the median age point should be provided."*

We have included in the text the $2\sigma$ range for climate events, corresponding to the maximal range we calculate from our chronology, itself mainly based on radiocarbon dates and especially for the onset of the Last Deglaciation and the Holocene.

*"The authors describe changes in assemblages or species that would suggest an un "atypical" climate changes that includes warm and wet conditions during the LGM and coldest reconstructed temperature of the record during the late Holocene. In indeed, these inferences are in contradiction with several (mostly geochemical) observations. I suggest therefore, a critical discussion about the robustness and weakness of the interpretation of the assemblage changes"*

Paleoecological interpretations in our study area are largely based on the modern dinocyst atlas (Zonneveld et al., 2013). In the revised manuscript, the ecology of main species discussed for the LGM atypical signature is highlighted thanks to maps highlighting the present-day distribution of taxa percentages in modern sediments: new Figure 6 inserted in the manuscript. Also, in order to be clearer, we have added a more critical discussion regarding dinocyst species and in particular for the thermophile ones that indeed lacked some present-day ecological information.

[Figure]

New Figure 6: Present-day distribution of selected dinocyst taxa among major ones discussed in the paper. Percentages from 277 sites are extracted from the modern dinocyst atlas (Marret et al., 2008; Zonneveld et al., 2013).

*"The authors need also to provide a more nuanced discussion about the mechanisms that control that the Benguela advection over their core location. In Figure 7, it is suggested that the Benguela advection was severely weakened during 4-5 ka and 7-15 ka BP. How does these interpretations compare with paleclimate records from more southern locations relative to that of the cores described in this study."*

In order to discuss the link between *Operculodinium centrocarpum* abundances and the influence of the Benguela Current over our study area, as mentioned in Zonneveld et al. (2013), a map of the actual geographical distribution of this species has been produced (cf. new Figure 6). Today, this species lives in the Benguela upwelling system and along the Angolan and Congolese margin until the equator (new Figure 6). The presence of *O. centrocarpum* north of the Angola-Benguela Front has been interpreted in the manuscript as the northward advection of cold waters during the austral winter, migrating westward around 1°S. This is illustrated below with satellite SST data of austral winter (green: 16°C, brown : above 27 °C) and the present-day abundances of *O. centrocarpum* percentages.

[Figure]

*Map of mean July SST in South Atlantic Ocean. Blue arrows display the major northward cold currents and yellow dashed line represents the Angola-Benguela Front.*

[Figure]

cf. new Figure 6: Present-day distribution of selected dinocyst taxa among major ones discussed in the paper. Percentages from 277 sites are extracted from the modern dinocyst atlas (Marret et al., 2008; Zonneveld et al., 2013).

Present-day occurrences of this dinocyst species appear to be consistent with the winter pattern of the Eastern South Subtropical Gyre (Benguela Current, South Atlantic Current and southern South Equatorial Current), but also with the Brazilian margin through the Falkland Current. This species then appears to be greatly correlated with northward cold currents.

We did not find any publications relative to the paleo-intensity of the Benguela Current. Modern surveys exist but are rather based on ENSO or Aguilhas Current monitoring. However, planktonic sediment traps allow discussing the actual relationship between dinocyst assemblages and current sea-surface conditions (Holzwarth et al., 2000; Zonneveld et al., 2001), showing that *O. centrocarpum* is the dominant species under the Bengula Current influence.

Paleo reconstructions mainly concern latitudinal shifts of the Angola-Benguela Front (Jansen et al., 1996; Uliana et al., 2002). We thus have added the Jansen et al. (1996) reconstructions in new Figure 8.

Concerning dinocyst-based studies in southern locations, there is GeoB 1016 core published by Dupont and Behling (2006), but the resolution is low regarding our chronological interval. There is finally the GeoB1023 core (Shi et al., 1998), a high resolution study carried out in the Benguela Upwelling System. Shi et al. (1998) did not discuss the Benguela Current activity (neither *O. centrocarpum* abundances). We can however extract *O. centrocarpum* abundances from this study and add this data in new Figure 8 to make regional comparisons.

*"A significant part of the Congo Basin is located north of the equator and, therefore, it is a part of the West African monsoon system. I highly encourage the authors to graphically compare their time-series of thermophile and low salinity assemblage/species with the highly resolved runoff and SST records from the Gulf of Guinea (Weldeab et al, 2007a-b, Weldeab et al., 2012a, Weldeab et al., 2012b). It is important that C2 BGD Interactive comment Printer-friendly version Discussion paper the authors' micro-paleontolagical approach is compared with and tested against the geochemical proxy records from the same region (see above reference)"*

We totally agree and we have added the Ba/Ca signal in Figure 6.

*There are several references cited in the main text but no listed in the reference list. It mismatch can avoided by using one of the several citation software and a careful checking.*

We fixed it.

*Wordings: there are several wordings (admitted, thanks to, climateuring, mitigation, tierce) that the authors need to replace with more appropriated words/phrases. Please change: "Weldeab et al" instead "Syee Weldeab et al.. "biogenic opal" instead "Biogenic Silica (BiSiO2)"*

We fixed it.

**REFERENCES OF PAPERS CITED IN THIS LETTER**

Dupont, L. and Behling, H.: Land–sea linkages during deglaciation: High-resolution records from the eastern Atlantic off the coast of Namibia and Angola (ODP site 1078), Quaternary International, 148(1), 19–28, doi:10.1016/j.quaint.2005.11.004, 2006.

Holzwarth, U., Esper, O. and Zonneveld, K.: Distribution of organic-walled dinoflagellate cysts in shelf surface sediments of the Benguela upwelling system in relationship to environmental conditions, Marine Micropaleontology, 64(1–2), 91–119, doi:10.1016/j.marmicro.2007.04.001, 2007.

Jansen, J. H. F., Ufkes, E. and Schneider, R. R.: Late Quaternary Movements of the Angola-Benguela Front, SE Atlantic, and Implications for Advection in the Equatorial Ocean, in The South Atlantic, pp. 553–575, Springer Berlin Heidelberg. [online] Available from: http://scdproxy.univ-brest.fr:2068/chapter/10.1007/978-3-642-80353-6_28 (Accessed 28 October 2014), 1996.

Shi, N., Dupont, L. M., Beug, H.-J. and Schneider, R.: Vegetation and climate changes during the last 21 000 years in S.W. Africa based on a marine pollen record, Veget Hist Archaebot, 7(3), 127–140, doi:10.1007/BF01374001, 1998.

Weldeab, S.: Bipolar modulation of millennial-scale West African monsoon variability during the last glacial (75,000–25,000 years ago), Quaternary Science Reviews, 40, 21–29, doi:10.1016/j.quascirev.2012.02.014, 2012.

Zonneveld, K.A.F., Hoek, R.P., Brinkhuis, H.,Willems, H., Geographical distributions of organic-walled dinoflagellate cysts in surficial sediments of the Benguela upwelling region and their relationship to upper ocean conditions, Progress in Oceanography, 48, 25-72, 2001.

Zonneveld, K. A. F., Marret, F., Versteegh, G. J. M., Bogus, K., Bonnet, S., Bouimetarhan, I., Crouch, E., de Vernal, A., Elshanawany, R., Edwards, L., Esper, O., Forke, S., Grøsfjeld, K., Henry, M., Holzwarth, U., Kielt, J.-F., Kim, S.-Y., Ladouceur, S., Ledu, D., Chen, L., Limoges, A., Londeix, L., Lu, S.-H., Mahmoud, M. S., Marino, G., Matsouka, K., Matthiessen, J., Mildenhal, D. C., Mudie, P., Neil, H. L., Pospelova, V., Qi, Y., Radi, T., Richerol, T., Rochon, A., Sangiorgi, F., Solignac, S., Turon, J.-L., Verleye, T., Wang, Y., Wang, Z. and Young, M.: Atlas of modern dinoflagellate cyst distribution based on 2405 data points, Review of Palaeobotany and Palynology, 191, 1–197, doi:10.1016/j.revpalbo.2012.08.003, 2013.

---

## Author Comment (AC2) · 6 Jul 2016

Dear reviewer,

We have corrected the manuscript taking into account the reviewer's suggestions and we have included all modifications in the revised manuscript. Below you will find our detailed responses to each of the reviewer's comments.

COMMENTS :

"Age model: I think it is critical that the authors perform a rigorous error analysis of their age model. I recommend age model uncertainty analysis for every data point using their tie-points and Bayesian statistical error analysis using softwares like "Bacon"

(Blaauw and Christen, 2011) or "B-Chron" (Parnell et al., 2008)"

We have plotted the 2-sigma range error in grayscale envelop around mean calibrated ages. The grey band plotted on the Age - Depth figure (first version) only corresponds to the 14C error (See attached figure 1)

"When discussing the onset or termination of an event, the median age of onset/termination of event and the uncertainty of the median age point should be provided."

We have included in the text the 2-sigma range for climate events, corresponding to the maximal range we calculate from our chronology, itself mainly based on radiocarbon dates and especially for the onset of the Last Deglaciation and the Holocene.

"The authors describe changes in assemblages or species that would suggest an un "atypical" climate changes that includes warm and wet conditions during the LGM and coldest reconstructed temperature of the record during the late Holocene. In indeed, these inferences are in contradiction with several (mostly geochemical) observations. I suggest therefore, a critical discussion about the robustness and weakness of the interpretation of the assemblage changes"

Paleoecological interpretations in our study area are largely based on the modern dinocyst atlas (Zonneveld et al., 2013). In the revised manuscript, the ecology of main species discussed for the LGM atypical signature is highlighted thanks to maps highlighting the present-day distribution of taxa percentages in modern sediments: new Figure 6 inserted in the manuscript. Also, in order to be clearer, we have added a more critical discussion regarding dinocyst species and in particular for the thermophile ones that indeed lacked some present-day ecological information. (See attached Figure 2)

"The authors need also to provide a more nuanced discussion about the mechanisms that control that the Benguela advection over their core location. In Figure 7, it is suggested that the Benguela advection was severely weakened during 4-5 ka and 7-

15 ka BP. How does these interpretations compare with paleclimate records from more southern locations relative to that of the cores described in this study."

In order to discuss the link between Operculodinium centrocarpum abundances and the influence of the Benguela Current over our study area, as mentioned in Zonneveld et al. (2013), a map of the actual geographical distribution of this species has been produced (cf. new Figure 6). Today, this species lives in the Benguela upwelling system and along the Angolan and Congolese margin until the equator (new Figure 6). The presence of O. centrocarpum north of the Angola-Benguela Front has been interpreted in the manuscript as the northward advection of cold waters during the austral winter, migrating westward around 1°S. This is illustrated below with satellite SST data of austral winter (green: 16°C, brown : above 27 °C) and the present-day abundances of O. centrocarpum percentages.

Present-day occurrences of this dinocyst species appear to be consistent with the winter pattern of the Eastern South Subtropical Gyre (Benguela Current, South Atlantic Current and southern South Equatorial Current), but also with the Brazilian margin through the Falkland Current (see attached Figure 3).

This species then appears to be greatly correlated with northward cold currents. We did not find any publications relative to the paleo-intensity of the Benguela Current. Modern surveys exist but are rather based on ENSO or Aguilhas Current monitoring. However, planktonic sediment traps allow discussing the actual relationship between dinocyst assemblages and current sea-surface conditions (Holzwarth et al., 2000; Zonneveld et al., 2001), showing that O. centrocarpum is the dominant species under the Bengula Current influence.

Paleo reconstructions mainly concern latitudinal shifts of the Angola-Benguela Front (Jansen et al., 1996; Uliana et al., 2002). We thus have added the Jansen et al. (1996) reconstructions in new Figure 8.

Concerning dinocyst-based studies in southern locations, there is GeoB 1016 core

published by Dupont and Behling (2006), but the resolution is low regarding our chronological interval. There is finally the GeoB1023 core (Shi et al., 1998), a high resolution study carried out in the Benguela Upwelling System. Shi et al. (1998) did not discuss the Benguela Current activity (neither O. centrocarpum abundances).

We can however extract O. centrocarpum abundances from this study and add this data in new Figure 8 to make regional comparisons.

"A significant part of the Congo Basin is located north of the equator and, therefore, it is a part of the West African monsoon system. I highly encourage the authors to graphically compare their time-series of thermophile and low salinity assemblage/species with the highly resolved runoff and SST records from the Gulf of Guinea (Weldeab et al, 2007a-b, Weldeab et al., 2012a, Weldeab et al., 2012b). It is important that C2 BGD Interactive comment Printer-friendly version Discussion paper the authors' micropaleontolagical approach is compared with and tested against the geochemical proxy records from the same region (see above reference)"

We totally agree and we have added the Ba/Ca signal in Figure 6 (now named Figure 7)

There are several references cited in the main text but no listed in the reference list. It mismatch can avoided by using one of the several citation software and a careful checking.

We fixed it.

Wordings: there are several wordings (admitted, thanks to, climateuring, mitigation, tierce) that the authors need to replace with more appropriated words/phrases. Please change: "Weldeab et al" instead "Syee Weldeab et al.. "biogenic opal" instead "Biogenic Silica (BiSiO2)"

We fixed it.

Sincerely yours,

William Hardy and co-authors

REFERENCES OF PAPERS CITED IN THIS LETTER

Dupont, L. and Behling, H.: Land–sea linkages during deglaciation: High-resolution records from the eastern Atlantic off the coast of Namibia and Angola (ODP site 1078), Quaternary International, 148(1), 19–28, doi:10.1016/j.quaint.2005.11.004, 2006.

Holzwarth, U., Esper, O. and Zonneveld, K.: Distribution of organic-walled dinoflagellate cysts in shelf surface sediments of the Benguela upwelling system in relationship to environmental conditions, Marine Micropaleontology, 64(1–2), 91–119, doi:10.1016/j.marmicro.2007.04.001, 2007.

Jansen, J. H. F., Ufkes, E. and Schneider, R. R.: Late Quaternary Movements of the Angola-Benguela Front, SE Atlantic, and Implications for Advection in the Equatorial Ocean, in The South Atlantic, pp. 553–575, Springer Berlin Heidelberg. [online] Available from: http://scdproxy.univ-brest.fr:2068/chapter/10.1007/978-3-642-80353-6_28 (Accessed 28 October 2014), 1996.

Shi, N., Dupont, L. M., Beug, H.-J. and Schneider, R.: Vegetation and climate changes during the last 21 000 years in S.W. Africa based on a marine pollen record, Veget Hist Archaebot, 7(3), 127–140, doi:10.1007/BF01374001, 1998.

Weldeab, S.: Bipolar modulation of millennial-scale West African monsoon variability during the last glacial (75,000–25,000 years ago), Quaternary Science Reviews, 40, 21–29, doi:10.1016/j.quascirev.2012.02.014, 2012.

Zonneveld, K.A.F., Hoek, R.P., Brinkhuis, H.,Willems, H., Geographical distributions of organic-walled dinoflagellate cysts in surficial sediments of the Benguela upwelling region and their relationship to upper ocean conditions, Progress in Oceanography, 48, 25-72, 2001.

Zonneveld, K. A. F., Marret, F., Versteegh, G. J. M., Bogus, K., Bonnet, S., Bouimetarhan, I., Crouch, E., de Vernal, A., Elshanawany, R., Edwards, L., Esper, O.,

Forke, S., Grøsfjeld, K., Henry, M., Holzwarth, U., Kielt, J.-F., Kim, S.-Y., Ladouceur, S., Ledu, D., Chen, L., Limoges, A., Londeix, L., Lu, S.-H., Mahmoud, M. S., Marino, G., Matsouka, K., Matthiessen, J., Mildenhal, D. C., Mudie, P., Neil, H. L., Pospelova, V., Qi, Y., Radi, T., Richerol, T., Rochon, A., Sangiorgi, F., Solignac, S., Turon, J.-L., Verleye, T., Wang, Y., Wang, Z. and Young, M.: Atlas of modern dinoflagellate cyst distribution based on 2405 data points, Review of Palaeobotany and Palynology, 191, 1–197, doi:10.1016/j.revpalbo.2012.08.003, 2013.

Please also note the supplement to this comment:
http://www.biogeosciences-discuss.net/bg-2016-148/bg-2016-148-AC2-supplement.pdf
* * *
[Figure]

Fig. 1. Figure 1 : fixed Figure 2 of the submitted paper. The grey band correspond now to the 2-sigma range error.

Lingulodinium machaerophorum          Spiniferites mirabilis          Spiniferites membranaceus

0% 10% 50% 80%          0%    10%    25%          0%    10%    25%

Tuberculodinium vancampoae          Operculodinium aguinawense          Operculodinium centrocarpum

0%   1%      6%          0%    10%    30%          0% 10% 50% 80%

**Fig. 2.** Figure 2 : Modern-day dinocyst location according modern Atlas (Zonneveld et al., 2013)

[Figure]

**Fig. 3.** Figure 3 : Map of mean July SST in South Atlantic Ocean. Blue arrows display the major northward cold currents and yellow dashed line represents the Angola-Benguela Front. Green : 16°C, brown : 27°C

[Figure]

**Fig. 4.** Figure 4 : Fixed Figure 8 according reviewer comments.

[Figure]

**Fig. 5.** Figure 5 : Firstly submitted version of Figure 8

---

## Author Response (AR2)

**Dear Editor,**

**Please find here our responses of your comments. Given that we have not any native English people in co-authors, we accepted most of your proposed corrections.**

**The corrections are displayed in red in the manuscript.**

**Best regards**

**William Hardy and co-authors**

**Comments accepted :**

line 77: "largest" missing after second

line148: change "added in" to "added to"

line 168-171: change to "….Marret et al.2008). A detailed discussion of limitations and pitfalls of inferring paleoproductivity from dinocyst assemblages in the study area will be discussed in a separate paper. In the present study, we only focus on…."

line189: need to close the bracket after "Figure 3"

line200: suggested wording: "Higher abundances of heterotrophic cysts, mainly…"

line240: "across the whole…"

line251: change to "….the southernmost occurrences ever recorded of this species is observed in core KZAI-01."

line271/272: suggested wording: "It is commonly accepted that in the intertropical band higher primary productivity occurred during periods of global colling…."

line280: move bracket to after core name, i.e. "core GeoB1008 (Schneider et al.; Fig)"

line282: delete "around"

line290/291: change "brings" to "leads"

line294: sensitive cysts PLURAL

line299: delete "then" before "also"

line303: change "forcing in" to "forcing on"

line309: change "it was evidenced" to "it has been suggested" or "it has been inferred"

line310: change "influence in" to "influence on"

line314: change "widespread" to "spread"

line334: delete "consequently"

line335: change "supposedly" to "supposed"

line337: delete "and" before "especially"

line345/346: change "This general and commonly admitted" to "This widely observed pattern"

line357: change "consistently" to "consistent"

line358-360 suggested wording: "Even if an influence of lower sea level on the neritic ecology of both E.spp and L.m. during this time cannot be ruled out entirely, this pattern…"

line363: change to "contribute half of the total discharge"

line364: change "underline" to "note"

line373: change "moisture within" to "moisture to"

line376: change "then" to "thus"

line377, 380: "consistently" to "consistent"

line384: change "scale" to "area" to "expanse"

line387: change "displaying" to "delineating"

line390: change "through" to "from"

line396: change wording to "…,when relative abundances of river-plume taxa increase"

line398/399: change to "…the maximum of West African monsoon activity between 8 and 6 ka BP (Figure 7) also corresponds…"

line407/408: change to "This pattern is well correlated with estimates of paleo precipitation inferred from pollen extracted from Burundi mounts"

line415/416: change to "Between 18 +/-0.3 and 15.5 +/- 0.4 ka BP, relative abundances of thermophile and river-plume species dropped sharply, while…"

line420: change "then" to "thus"

line425: change to "which today is typical…"

line429: precipitation SINGULAR

line431: "mean LGM values"

line433: delete "melting"

line435: change to "….would then consist of a southward shift…."

line439: change "all along" to "during"

line447: delete "activities"

line448/449: change to "Our dinocyst data also show a significant increase in percentages of XX and XX at around 15.5 +/-0.4kaBP"

line450: change to "…in the tropics are related to both nutrient-enriched…."

line453: delete "study"

line454: change "occurred" to "appeared"

line455: change to "Relative abundances of L.m…. also increased…"

line461: delete "as"

line468: delete "then"

line474: change to "evidences a reduction of terrigenous input (SINGULAR)…."

line475: delete "then"

line479: change to "….at that time (Figure 8). Wetter tropical conditions….."

line488: change "well abundant" to "highly abundant" or just "abundant"

line492-494: change to "Acknowledging the weak chronostratigraphic constraints (cf Figure2), some major subdivisions of the Holocene (xxx) can be discussed nevertheless"

line505: change to "Similar to these studies, our record…."

line512: delete "also"

line513: "relative" not in CAPS

line519: change to "The timing of the AHP termination varies significantly between studies, i.e. around 2.5……"

line523/524: change to "Consequently, the duration of the AHP termination also varies between these studies, from a few centuries to a few thousand years."

line530: change to "…from eutrophic to less nutrient rich surface waters…."

line550: delete "ever"

line551: delete "within dinocyst assemblages"

line561: delete "reached"

line572: change to "Analysis of dinocyst assemblages in core KZAI-01 has permitted…"

line574/575: change to "…river discharges caused by latitudinal migrations of the tropical rainbelt, forced by different orbital configurations,…"

line577: delete "tropical"

line579: delete "latitudinal"

line581: change to "complex dynamics that warrant model simulations to explore the underlying mechanisms that occurred…."

line588: change to "…in relationship with latitudinal shifts in the tropical rainbelt…"

**Discussed comments :**

line 271/272: since you state "commonly accepted", you need to provide references here

The references were provided below at line 276-277. We moved them to the line 271-272.

line525/526: I cannot understand what is meant by "degradation steps" and by "heat and moisture decrease". Does the latter mean cooling and drying?

We would explain that the establishment of drier and cooler conditions after Early Holocene was not gradual but established abruptly in two steps : the first at 7 ka BP and the second at 4 ka BP. These recorded transitions appear to be abrupt, less than 500 years.

We change the wording to « **In our data, we observe two events characterised by abrupt cooling and drying conditions."**

line566: not sure what "mitigated" means here

We would say an intermediate period which still was warm and wet but not as warm/wet as Early Holocene.

We propose to replace "mitigated" by "less"

line578: what is this "expectation" based on?

We changed to « **Furthermore, while most of studies describe the LGM as a "cold and dry" period in the tropics, thermophile and river-plume dinocysts evidence here a pattern relatively similar to modern warm equatorial assemblages.** »